# ACE2 and Innate Immunity in the Regulation of SARS-CoV-2-Induced Acute Lung Injury: A Review

**DOI:** 10.3390/ijms222111483

**Published:** 2021-10-25

**Authors:** Lihua Qu, Chao Chen, Tong Yin, Qian Fang, Zizhan Hong, Rui Zhou, Hongbin Tang, Huifen Dong

**Affiliations:** 1Department of Pathogenic Biology, School of Basic Medical Sciences, Wuhan University, Wuhan 430071, China; lihuaqu@whu.edu.cn (L.Q.); yintong@whu.edu.cn (T.Y.); qianfang@whu.edu.cn (Q.F.); zizhanhong@whu.edu.cn (Z.H.); ruizhou@whu.edu.cn (R.Z.); 2School of Medicine & Holistic Integrative Medicine, Nanjing University of Chinese Medicine, Nanjing 210013, China; 20193068@njucm.edu.cn; 3Center for Animal Experiment, State Key Laboratory of Virology, Wuhan University, Wuhan 430071, China

**Keywords:** ACE2, immune cells, COVID-19, SARS-CoV-2, ALI

## Abstract

Despite the protracted battle against coronavirus acute respiratory infection (COVID-19) and the rapid evolution of the severe acute respiratory syndrome coronavirus 2 (SARS-CoV-2), no specific and effective drugs have to date been reported. Angiotensin-converting enzyme 2 (ACE2) is a zinc metalloproteinase and a critical modulator of the renin-angiotensin system (RAS). In addition, ACE2 has anti-inflammatory and antifibrosis functions. ACE has become widely known in the past decade as it has been identified as the primary receptor for SARS-CoV and SARS-CoV-2, being closely associated with their infection. SARS-CoV-2 primarily targets the lung, which induces a cytokine storm by infecting alveolar cells, resulting in tissue damage and eventually severe acute respiratory syndrome. In the lung, innate immunity acts as a critical line of defense against pathogens, including SARS-CoV-2. This review aims to summarize the regulation of ACE2, and lung host cells resist SARS-CoV-2 invasion by activating innate immunity response. Finally, we discuss ACE2 as a therapeutic target, providing reference and enlightenment for the clinical treatment of COVID-19.

## 1. Introduction

Coronavirus acute respiratory infection 2019 (COVID-19) induced by severe acute respiratory syndrome coronavirus 2 (SARS-CoV-2) is a worldwide acute respiratory disease. SARS-CoV-2 has higher infectivity than Middle East respiratory syndrome coronavirus (MERS-CoV) and severe acute respiratory syndrome coronavirus (SARS-CoV) [1,2,3]. The pathological features of COVID-19 are similar to SARS and MERS, including extensive edema, hyaline membrane formation, inflammatory infiltration, micro-thrombosis, and fibrosis [4,5]. Currently, there are more than 185 million COVID-19 cases worldwide, 10–20% of which have manifested acute lung injury (ALI) and even developed acute respiratory distress syndrome (ARDS), with an associated mortality of about 3% [1,6,7,8]. ALI is characterized by increased permeability of the pulmonary capillary and numerous immune cells, such as macrophages, neutrophils, lymphocytes, and dendritic cells (DCs), participate in the inflammatory response against SARS-CoV-2. A potent response against the virus can induce a cytokine storm and lead to ALI [9,10,11,12]. 

Angiotensin-converting enzyme 2 (ACE2) is a homolog of angiotensin-converting enzyme (ACE) expressed in human lungs and immune cells. ACE2 is known to antagonize the renin-angiotensin system (RAS) to alleviate ALI [13]. After the SARS outbreak in 2003, the critical role of ACE2 in ALI/ARDS has attracted widespread attention. Clinical studies showed that the insertion or deletion of ACE could impact the severity of ARDS [14,15]. Studies have reported that SARS-CoV2 has highly homologous sequences with SARS-CoV, which binds to ACE2, causing ALI by using spikes [16,17]. ACE2 and pattern recognition receptors are molecules expressed by innate immune cells. This review discusses the role of ACE2 and innate immunity in SARS-CoV2, which provide novel therapies for the prevention and control of SARS-CoV-2.

## 2. ACE2 and COVID-19

### 2.1. The Biological Function of ACE2

ACE2, and ACE homolog, belongs to the zinc metalloprotease family [18,19,20]. The *ACE2* gene is localized on the X chromosome with 18 exons, and it encodes a type I transmembrane glycoprotein composed of 805 amino acids, which includes an extracellular catalytic domain at the N-terminus and an intracellular sequence of 42 amino acids at the C-terminus [21,22]. The active site of the catalytic domain, the zinc metallopeptidase domain, shares 41.8% sequence identity with ACE. Although, there is some similarity, they catalyze different substrates and have different biological functions [23,24]. RAS is one of the essential vasoactive systems and plays a critical role in maintaining blood pressure and fluid–electrolyte balance through endocrine, paracrine, and autocrine systems [25]. In addition, it has been reported that RAS is involved in many inflammation-related pathological reactions [26,27,28]. RAS consists of several functionally interacting protease–hormone-receptor axes [29,30,31]. Among them, the ACE/angiotensin II (Ang II)/angiotensin type 1 receptor (AT1R) pathway is associated with cardiovascular fibrosis, oxidative stress, inflammation, cell apoptosis, proliferation, and cellular immunity [32,33]. ACE2 is a central player in the ACE2/Ang (1-9)/AT2R and ACE2/Ang- (1-7)/MasR axes [34]. ACE2 catalyzes the hydrolysis of angiotensin I (Ang I) to the nonapeptide Ang (1-9) and Ang II to the heptapeptide Ang (1-7), respectively [35]. Ang (1-9) and Ang (1-7) mainly bind to their corresponding G-protein-coupled receptors, AT2R, or Mas receptor (MasR), and exert biological effects of vasodilation, antifibrosis, anti-inflammatory, and immune cell activation [36,37] (Figure 1). ACE2 has a higher affinity for Ang II than Ang I, and the former works as its primary substrate [38,39,40].

### 2.2. ACE2 and SARS-CoV-2 in ALI

Audrey et al. [41] explored mice as a virus infection model, and found that SARS-CoV-2 could not replicate in wild-type mice. However, severe ALI occurred in mice by adapting the virus to mouse ACE2 receptor and inserting the human ACE2 gene. Zhou et al. [42] found that coronavirus entered the host cells by utilizing the spike protein to bind the hydrophobic pocket of the ACE2 extracellular catalytic domain. Once infected by the virus, ACE2 was downregulated in the cell and thus led to a dysregulation of the ACE2/Ang- (1-7)/MasR axis and the ACE/Ang II/AT1R axis [43]. Consequently, Ang II was upregulated, and AT1R was overstimulated, increasing capillary permeability, and causing pulmonary edema and ALI [44]. Imai et al. [23] demonstrated that the ACE2/Ang (1-7)/MasR axis has protective properties in the lungs, relieving pulmonary inflammation and fibrosis and inhibiting cancer cell growth, tumor angiogenesis, and metastasis. ACE2 also inhibited pulmonary fibrosis and lung injury caused by overactivation of the RAS system in ALI mediated by SARS and influenza viruses [44,45]. On the other hand, RAS inhibitors could upregulate ACE2 and reduce lung injury [23,46]. For instance, ACE2 knockout mice following treatment with an AT1R inhibitor or recombinant human ACE2 manifested less coronavirus-induced ALI [47,48]. Chen et al.’s [49] studies showed that glycyrrhizic acid attenuated inflammatory factors and adhesion molecules by upregulating ACE2 and inhibiting the caveolin-1/NF-κB signaling pathway, thereby alleviating LPS-induced ALI. Similarly, research on the influenza virus suggested that ACE2 is significantly downregulated upon H1N1 infection [50]. ACE2 deficiency deteriorated the condition of infected mice significantly, and AT1R blockade alleviated the lung damage caused by the H7N9 influenza virus [51,52]. In addition, an elevated level of Ang-II was found in patients with H5N1 or H7N9 infection, and these elevated levels were related to the severity of the pulmonary injury [52].

Gerard et al. [53] investigated the expression and distribution of ACE2 in COVID-19 and showed that the expression of ACE2 increased in lung tissues, serum, and endothelial cells but decreased in alveolar epithelial cells (AT2). Asaka et al. [54] explored the pathogenicity of COVID-19 and found that CAG-hACE2 mice were susceptible to SARS-CoV-2 and the levels of cytokines and chemokines increased, resulting in acute lung injury, while injecting the plasma of immunized mice could reduce the mortality of mice. Recent studies reported that SARS-CoV-2 infection contributed to RAS dysregulation [55,56]. SARS-CoV-2 patients had a higher level of Ang II, and administration of ACE inhibitors (ACEIs) could significantly reduce cytokine production and pulmonary inflammatory responses [57,58]. Therefore, drugs balancing the RAS axis, such as ACEIs and angiotensin receptor blockers, might be suitable for treating patients with COVID-19 [59,60,61]. In addition, previous studies reported coronavirus E protein as the determinant of SARS-CoV pathogenicity [62], as E protein triggered overexpression of inflammatory cytokines and aggravated the immune response, resulting in ALI, and eventually ARDS [61,63,64]. Collectively, these results suggested that suppression of RAS and ACE2 was involved in the pathogenesis of COVID-19 lung injury.

## 3. The Role of ACE2 in Innate Immune-Related Cells during SARS-CoV-2 Infection

ACE2 is abundantly expressed in alveolar epithelial cells, endothelial cells, macrophages, neutrophils, DCs, and lymphocytes in lung tissue [65,66,67]. Therefore, lung tissues are especially vulnerable to SARS-CoV2 invasion and the main target organ for virus attack [68,69,70].

SARS-CoV-2 binds to ACE2 in type II alveolar cells after entering the respiratory tract, resulting in decreased ACE2 and increased blood Ang II [71]. The association of Ang II with AT1R could induce bronchial smooth muscle contraction, pulmonary vascular hyperpermeability, alveolar epithelial cell apoptosis, and the release of numerous inflammatory cytokines and chemokines, which led to ARDS [72,73]. ACE2 on alveolar macrophages could relieve lung tissue injury and inflammation through adenosine monophosphate-activated protein kinase (AMPK) and mammalian target of rapamycin (mTOR) pathways [74,75,76]. Besides, the ACE2/Ang- (1-7)/MasR axis reduced the secretion of proinflammatory factors and apoptosis of alveolar epithelial cells and vascular endothelial cells by inhibiting JNK/NF-κB activation [77]. However, blockage of the MasR pathway impaired the protective effect of Ang- (1-7) and aggravated ARDS [78].

Numerous immune cells are activated upon SARS-CoV-2 infection in the alveoli, which downregulates ACE2 and increases the expression of Ang II, proinflammatory IL-1β, IL-6, TNF-α, and the chemokines CXCL8, CXCL10 [79,80]. Ang II activates NF-κB and p38MAPK through the combination of AT1R, which produces numerous inflammatory factors, and in turn, activates the inflammatory response and triggers a massive accumulation of macrophages and neutrophils, aggravating lung injury [81]. Neutrophils, macrophages, and other immune cells are recruited to the lungs in the early stage of the disease to participate in the clearance of the virus [82]. Furthermore, viruses have a dynamic mutual interaction with alveolar epithelial cells and macrophages. The viruses induce the release of cytokines, such as monocyte chemotactic factor (MCP-1) and granulocyte-macrophage colony-stimulating factor (GM-CSF), by macrophages and neutrophils. Cytokine release activates macrophages and generates inflammatory factors, leading to lung inflammation and increased exudation. Alveolar expansion is also restricted, affecting the alveolar ventilation function, and eventually causing refractory hypoxemia and aggravating ALI [83,84]. Macrophages and neutrophils release IL-6, TNF-α, IL-1β, and CXCL5 to promote DCs’ maturation and the activation and migration of CD4+ and CD8+ T cells, which creates a microenvironment that is advantageous for immune cell migration and accumulation, activating the adaptive immune system [85]. In DCs, Ang-(1-7) stimulated ERK1/2 phosphorylation through AT2R, and inhibited the proinflammatory Ang II. Numerous inflammatory mediator factors, such as IL-1β, TNF-α, IL-6, TGF-β, and CXCL9, are produced and ultimately contribute to the cytokine storm and ALI [86,87]. Chen et al. [88] found that the spike protein of SARS-CoV-2 mediated viral entry through the binding to ACE2 on the cell surface. Menezes et al. [89] and Li et al. [90] hypothesized that immune cells, including monocytes, neutrophils, lymphocytes, and natural killer cells, could travel to infected areas after breaking endothelial and epithelial barriers. These immune cells could eliminate infected cells as well as alveolar exudates, leading to uncontrolled inflammation [89,90]. Patel et al. [91] found that patients who died from SARS-CoV-2 manifested massive infiltration of various immune cells in their alveolus, which formed a cytokine storm and resulted in lung failure (Figure 2).

## 4. ACE2, SARS-CoV-2, and Innate Immune Pathways

The association between the innate immune system and SARS-CoV-2 is crucial for regulating viral infection, disease progression, and prognosis [92,93]. Innate immunity relies on pattern recognition receptors (PRRs) by multiple cell surfaces or intracellular receptors and responds by signal transduction to effector molecules [94,95]. SARS-CoV-2 infection activates pattern recognition receptors on pulmonary epithelial cells, endothelial cells, macrophages, DCs, and other immune cells to produce cytokines [96,97]. Recognition receptors including Toll-like receptors (TLRs), retinoic acid-inducible gene I (RIG-I, melanoma differentiation-associated gene 5 (MDA5), cyclic guanosine phosphate adenosine phosphate synthase (cGAS), stimulator of interferon genes (STING), and nucleotide-binding oligomerization domain-like receptors (NLRs) play a significant role in antiviral defense against coronaviruses (Figure 3) [98,99,100].

### 4.1. TLRs Pathway

TLRs are a family of transmembrane receptors that are important in the immune system [101]. TLRs recognize SARS-CoV-2 and induce activation of the immune system and inflammation [98,102,103]. Elenad et al. [104] found that the ACE2/Ang-(1-7)/AT2R axis interact with TLRs to regulate the T lymphocyte response, thereby playing an anti-inflammatory role. TLRs have several family members, among which, TLR3, TLR7, and TLR8 are expressed on the surface of cellular endosomes, recognizing viruses as transmembrane receptors [105,106]. Xia et al. [107] found that Chinese herbal medicines show good affinity for SARS-CoV2 3CLpro and ACE2 and could inhibit tumor necrosis factor (TNF) and TLRs. Toll-like receptor 2 (TLR2) is a major pattern recognition receptor (PRR) and mediates viral infection via the TLR2/MyD88 pathway [108,109]. Dosch et al. [110] found that the S protein of SARS-CoV-2 could trigger activation and nucleus translocation of NF-κB and cause an inflammatory response in peripheral blood monocytes (PBMCs) through TLR2. Choudhury et al. [98] showed that S proteins of SARS-CoV-2 could also interact with hydrogen bonds and hydrophobic areas of TLR1, TLR4, and TLR6 besides ACE2. In addition, TLR4 initiated the key pathway to regulate ALI. Imai et al. [111] reported that mice infected with influenza virus hemagglutinin 5 neuraminidase 1 (H5N1), SARS, or *Bacillus anthracis* showed an excessive production of oxidized phospholipids (OxPAPC), which activated the TLR4–TRIIF–TRAF6 signaling pathway and released massive inflammatory factors from macrophages, inducing ALI. TLR4 knockdown impaired H5N1 infection in the lung and alleviated ALI [111]. TLR3 recognizes ligands to initiate the signaling through TIR-containing adaptor molecule (TICAM-1/TRIF), followed by the activation of NF-κB and interferon regulatory factor 3 (IRF3). These transcription factors modulate the release of type-I IFN from immune cells, mediating innate antiviral immunity [112,113]. Treatment with Poly I: C, a double-stranded RNA analog that interacts with TLR3, can directly activate alveolar macrophages and DCs, eliminating the inhibition of alveolar macrophages [114]. Moreover, Nakazono et al. [115] reported that Poly I:C could bind to TLR3 to activate IFN and the NF-κ B pathway, promoting SARS-CoV-2 entry into nasal mucosal epithelial cells to induce ACE2 expression. MyD88 (myeloid cell differentiation factor 88), a signal adaptor protein of TLR7 and TLR8, can recognize single-stranded ribonucleic acid (ssRNA) [116]. During SARS, MyD88 mediates the transmission of natural immune signals and recruits inflammatory cells to the lung and induces human plasma DCs (pDCs) to produce type-I IFN signaling through TLR7 to control rapid viral replication [117]. It was also found that SARS-CoV ssRNA had an intense immunological activity and could stimulate numerous IL-6, TNF-α, and IL-12 via TLR7 and TLR8 signaling pathways, which induced a cytokine storm in mice and finally caused ALI [118,119,120,121,122]. Studies have found that chemical compounds (PHA-408) could block TLRs and reduce inflammation [123]. In conclusion, blocking TLR7- and TLR8-mediated signaling pathways could inhibit the inflammatory response induced by a cytokine storm. However, whether this strategy could be applied for the treatment of coronavirus infection remains to be further studied.

### 4.2. RIG-I/MDA5 Pathway

RIG-I is a multidomain intracellular protein highly related to anti-RNA virus immunity [124,125]. The RIG-I/MDA5 signaling pathway mainly recognizes viral RNA in the cytoplasm and induces a natural immune response and inflammation, thereby controlling infection [126,127,128]. Studies have shown that SARS-CoV-2 RNA could trigger activation of the RIG-I-MAVS pathway and enhance the release of type-I IFN [129]. Yang et al. [130] found that after knocking out RIG-I, MDA5, or mitochondrial antiviral-signaling protein (MAVS), SARS-CoV-2-infected human endothelial cells produced notably less type-I/III IFN and the expression of ACE2 was increased. In alveolar epithelial cells, ORF9B inhibited the release of IFN, thereby inhibiting antiviral immunity [129]. Viral particles fuse with cellular membranes or vacuoles during their invasion to release dsRNA into the cytoplasm, and RIG-I/MDA5 can identify viral RNAs and catalyze their own activation [131]. Innate immunity is subsequently induced via downstream MAVS and IPS-1, activating multiple complexes, such as TRAF3, TANK, IKK, TBK1, and p-IRF3. This leads to the activation of NF-κB and IRF3, which participate in the transcription of type-I IFN and play antiviral functions in the initial stage of infection [112,132,133,134,135].

### 4.3. cGAS/STING Pathway

The cGAS/STING pathway stimulates type-I IFN during DNA and RNA viral infections, an essential factor in resisting viral infections [136,137]. DNA viruses, including adenovirus and vaccinia virus, can activate cGAS/STING, mice deficient in this pathway are more susceptive to viral infection [138]. During SARS-CoV-2 infection, cGAS binds with dsDNA to catalyze its transformation into cGAMP. cGAMP acts as a second messenger and it interacts with the adaptor protein STING on the endoplasmic reticulum (ER) membrane [139] to activate downstream kinase complex TANK-binding kinase 1 (TBK1) and inhibitor of κB protein kinase (IKK), resulting in nuclear translocation of IRF3 and NF-κB [140,141,142].

The cGAS/STING pathway protects host cells against RNA viruses. Fiachra et al. [143] found that in K18-hACE2-transfected mice with or without SARS-CoV-2 infection, intranasal administration of diazi-4 could activate STING, and inhibit viral replication in lung epithelial cells. Knockout of cGAS or STING genes in vitro or in vivo promoted large-scale replication of RNA viruses [144,145]. Applying STING agonist to influenza vaccine could improve its ability to resist multiple influenza viruses [146]. ORF3a of SARS-CoV-2 bound to STING and inhibited downstream signaling, restraining the nuclear accumulation of p65, preventing type-I IFN expression [137,147,148,149,150]. Berthelot et al. [151] found that the combination of SARS-CoV2 and ACE2 induced Ang II in mice, leading to overactivation of the STING pathway, which promoted tissue damage via monocyte-macrophage releasing interferon-β and tissue factors. cGAS-STING could not recognize viral RNA directly, but it could be activated by damaged host dsDNA, inducing the expression of type-I IFN and participating in the defense against SARS-CoV-2 infection with TLR3 and RIGI-MDA5 pathways [152,153,154].

### 4.4. NLRs Signaling Pathway

The NOD-like receptor (NLR) family is a family of immune receptors that identify pathogen-related cytoplasmic proteins. NOD-like receptor family pyrin domain containing 3 (NLRP3) is a component of the NLRs family, which plays a vital role in resisting viral infection in the lung by identifying viral RNA and subsequently activating the inflammasome [155]. Studies reported that SARS-CoV-2 activated NLRP3 after binding to ACE2 on vascular endothelial cells, inducing pyroptotic cell death, while the ACE2/Ang (1-7)/Mas axis regulated pyrolysis by inhibiting NLRP3 and eased the cytokine storm to exert a protective effect [156,157,158,159]. The NLRP3 inflammasome could be activated through the trans-Golgi network (TGN) owing to the stimulation of ion channels in the host cell membrane by viral porins. These viral porins, including the envelope (E) protein of the SARS-CoV2 virus and the M2 protein of the influenza virus, could subsequently activate caspase-1 [160,161,162]. NLRP3 activated by viral porins triggered pre-IL-1β and pre-IL-18 hydrolysis by caspase-1, generating excessive IL-1β, IL-18, and TNF-α and recruiting immune cells into the lung, leading to the death of infected cells. At the same time, reduced secretion of inflammatory factors was exhibited in NLRP3 knockout mice, which contributed to a higher mobility rate [155,160,163]. Studies have confirmed that patients infected with SARS and MERS had elevated levels of IL-1β and IL-18 in the lungs and lymphatic tissues, suggesting inflammasome activation [164,165]. Furthermore, the expression of TNF-α and IL-1β upon inflammasome complex activation promoted the accumulation of IL-6, which could act as one of the main impetuses of SARS-CoV-induced pulmonary inflammation and ARDS [160,166].

## 5. The Potential Therapeutic Role of ACE2 in COVID-19

Currently, there is no specific drug or vaccine for COVID-19. According to statistics, the mortality rate of severe COVID-19 patients is as high as 22–44% [72,167]. There is an urgent need for more effective treatments in the clinic. ACE2, as a binding protein of SARS-CoV2, not only plays a vital role in viral infection [168,169], but its dysfunction also leads to the weakness of the inflammatory inhibitory effect of ACE2/Ang-(1-7)/MASR axis, resulting in the aggravation of lung injury [170]. In addition, ACE2 is also a receptor protein for SARS-CoV-2-infected cells. Considering the potential treatment of ACE2-mediated COVID-19, some feasible treatments will be very important [171,172]. Therefore, ACE2 may play a certain therapeutic role on COVID-19 (Table 1).

### 5.1. Chloroquine

Studies have shown that chloroquine reduced viral infection by obstructing the binding of SARS-CoV-2 to ACE2 [173,174,175]. Gies et al. [172] found that chloroquine could destroy the terminal glycosylation of ACE2. This contributed to the conformational change of ACE2, therefore disturbing ACE2-SARS-CoV-2 binding and inhibiting viral replication in the host cells. Chloroquine also has anti-inflammatory and antiautophagy functions, which alleviate H5N1-induced lung injury in mice [176]. Therefore, chloroquine might have therapeutic effects on SARS. However, some studies have shown that chloroquine had no therapeutic effect on patients [177,178,179], and even had serious side effects and increased risk of death [180,181,182]. For these reasons, some hospitals have stopped using the drug.

### 5.2. ACEI or ARBs

Extensive attention has been attracted to whether ACEI/ARBs can be applied to the treatment of COVID-19 [183]. Neyrinck et al. [184] demonstrated that ACEI had a therapeutic effect on endotoxin-induced lung injury in rats. Melissa et al. [185] showed that ACEI and ARBs might improve ARDS by inhibiting the ACE/Ang II/AT1R signaling pathway. ARBs inhibited the canonical RAS pathway through angiotensin /Ang II/ATR1 axis; increasing ACE2, TLR-2, and IL-1β; and causing accumulation of reactive oxygen species (ROS), while ACE2 upregulation activates Ang-(1-7)/Mas pathway and inhibits inflammatory signals [186]. Studies identified that ARB could rescue SARS-CoV-2 spike or influenza virus-mediated ALI [187,188]. However, other studies inferred that there could be adverse aspects of using ACEI/ARB [189]. A recent study found that ACE2 was significantly upregulated after SARS-CoV and MERS-CoV infection, which could enhance their infection and transmission ability, and boosted the severity of COVID-19 [190]. However, ACEIs/ARBs were not associated with enhanced SARS-CoV-2 infection but led to decreased mortality [189,191]. Therefore, further investigations are still needed to demonstrate the function of ACEI/ARBs in COVID-19.

### 5.3. Recombinant ACE2

Recombinant ACE2, a recombinant soluble receptor that has a potential therapeutic effect on ALI, as it blocks the binding of SARS-CoV-2 to ACE2-expressing cells, thus avoiding viral infection [40,192,193,194,195]. Hoepel et al. [196] constructed recombinant proteins using an Fc fragment of human IgG1 together with the extracellular domain of either normal or mutant ACE2. Both recombinant proteins could interact with SARS-CoV-2 spike protein and therefore obstruct viral invasion. Monteil et al. [197] reported that human recombinant ACE2 (rhACE2) significantly prevented SARS-CoV-2 invasion, indicating that rhACE2 plays a role in the initial stage of the disease. However, whether this kind of drug is effective only for early infection remains uncertain [193]. Huang et al. [198] suggested that recombinant ACE2-Fc proteins not only worked as antibodies to block viral invasion and generate long-term immunity, but they could also act as a complement to decrease pulmonary ACE2 during infection, which improved the pathologic conditions of ARDS directly.

### 5.4. Vitamin D

Vitamin D has anti-inflammatory properties and plays a crucial role in immunity [199,200,201]. Li et al. [202,203] presented that 1,25(OH)_2_D3, the active form of vitamin D, also called calcitriol, was a negative regulator of RAS and inhibited renin synthesis. Xu et al. [204] confirmed that vitamin D could block ACE and Ang II expression and the increase of the ACE2 level in LPS-induced ALI. Mendonca et al. [205] found that activation of Nrf2 by vitamin D could reduce oxidative stress and inflammation, enhance innate immunity, and downregulate ACE2 to reduce the severity of SARS-CoV-2 infection.

**Table 1 ijms-22-11483-t001:** The potential therapeutic role of ACE2 in SARS-CoV-2-induced acute lung injury.

Drug	Major Outcome(s) Relates to ACE2 and SARS-CoV-2	References
Chloroquine	Chloroquine inhibited the binding of SARS-CoV-2 to ACE2, reducing the infection of the host cell by the virus. However, an increase in overall mortality was found in patients treated with chloroquine.	Ortiz MEetal., 2020 [173] Devaux et al., 2020 [174] Joseph et al., 2020 [182]
ACEI or ARBs	ACEI or ARBs inhibited the ACE/Ang/AT1R pathway to reduce inflammatory response and alleviate ARDS.	Neyrinck et al., 2009 [184] Melissa et al., 2021 [185] Yisireyili M et al., 2018 [186]
rhACE2	rhACE2 could bind to the spike protein by competing with ACE2 on the cell membrane surface, which on the one hand, inhibited the virus from infecting cells, on the other hand, rhACE2 could activate ACE2-Ang (1-7)-MasR pathway, reducing lung inflammation, and alleviating lung injury or ARDS.	Gheblawi et al., 2020 [40] Guzik et al., 2020 [192] Hoepel et al., 2021 [196]
Vitamin D	Vitamin D reduced oxidative stress and inflammation, enhancing innate immunity, and downregulating the expression of ACE2 to reduce the severity of SARS-CoV-2 infection.	Mendonca et al., 2020 [205]

## 6. Conclusions Remarks and Future Perspectives

In summary, RAS and ACE2/Ang- (1-7)/MasR axes play a critical role in ALI induced by SARS-CoV-2. Normal cellular immune responses have important protection against SARS-CoV-2 in lung host cells, but overexcited immune responses could induce cytokine storms, leading to cell death. This will eventually cause damage to the lung tissues and affect the normal respiratory function of the lung. Although SARS-CoV-2 and SARS-CoV are highly homologous, there are different characteristics between SARS-CoV-2, MERS-CoV, SARS-CoV, Ebola virus, and influenza A virus H1N1 infection. SARS-CoV-2 and SARS-CoV bind to ACE2 on the cell membrane using their spike protein. This mediates viral entry into cells, and on the other hand, may also destroy ACE2, resulting in an imbalance of RAS and ACE2/Ang- (1-7)/MasR. This dysregulation can exacerbate the disease and lead to ALI. Overreactive immune cells can generate cytokine storms, deteriorating multiple organs functions throughout the body. Therefore, blocking the spike binding to ACE2 or restoring the balance of RAS and ACE2/Ang-(1-7)/MasR are potential targets for developing specific drugs, antibodies, and vaccines for COVID-19 treatment. More studies on SARS-CoV and ACE2 are needed to understand the pathological mechanism of SARS-SoV-2 and to develop new preventive and treatment tools for COVID-19 in an effort to overcome the pandemic.

## Figures and Tables

**Figure 1 ijms-22-11483-f001:**
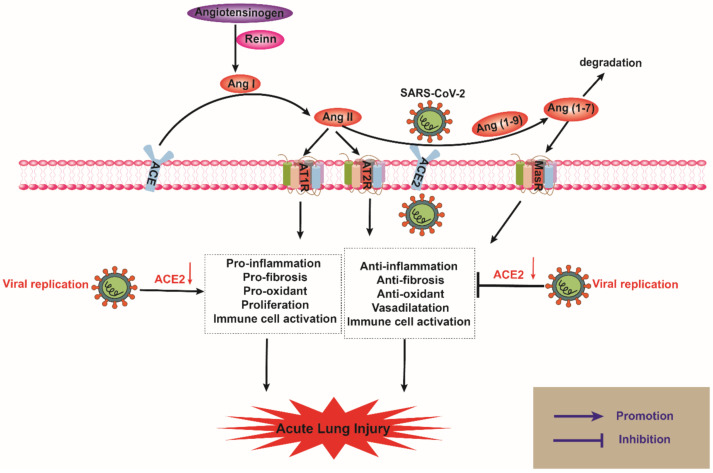
Current view on the renin angiotensin system (RAS) and development of ALI in COVID-19. RAS consists of classical and non-classical axes. ACE is a component of the classical axis, which converts angiotensin I (Ang-I) to Ang-II, and the latter binds to AT1 or AT2 receptors. The non-classical RAS axis contains ACE2, which hydrolyzes Ang II to produce Ang-(1-7). Ang-(1-7) has affinity with the Mas receptor (MasR), and they altogether form the ACE2/Ang-(1-7)/MasR axis, which can regulate the ACE/Ang II/AT1R axis. When lung tissues are infected with SARS-CoV-2, the virus binds with ACE2 receptor on the cell membrane and downregulates its expression, causing Ang II to activate the AT1 receptor and promote ALI. The red arrow indicates a reduction.

**Figure 2 ijms-22-11483-f002:**
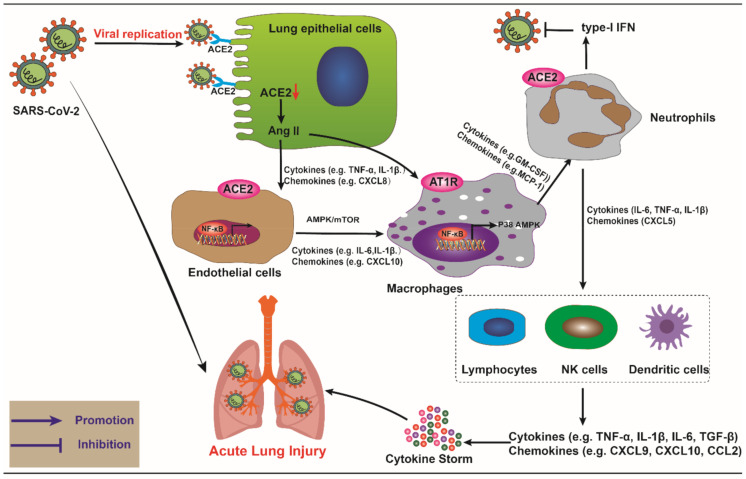
The cellular immune response of ALI induced by SARS-CoV2 infection. SARS-CoV-2 binds to ACE2 in type II alveolar cells after entering the respiratory tract, resulting in decreased ACE2 and increased Ang II. The association of Ang II with AT1R could induce bronchial smooth muscle contraction, pulmonary vascular hyperpermeability, alveolar epithelial release of numerous inflammatory cytokines, and chemokines. Alveolar macrophages could relieve lung tissue injury and inflammation through adenosine monophosphate-activated protein kinase (AMPK) and mammalian target of rapamycin (mTOR) pathways. Numerous cytokines and chemokines activate neutrophils to produce type-I IFN, inhibit virus infection, and neutrophils release numerous inflammatory factors and chemokines, promote the maturation of natural killer cells (NKs), dendritic cells (DCs), and the activation and migration of lymphocytes, activate the adaptive immune system, leading to further expansion of the immune response, and ultimately triggering a cytokine storm. The red arrow indicates a reduction.

**Figure 3 ijms-22-11483-f003:**
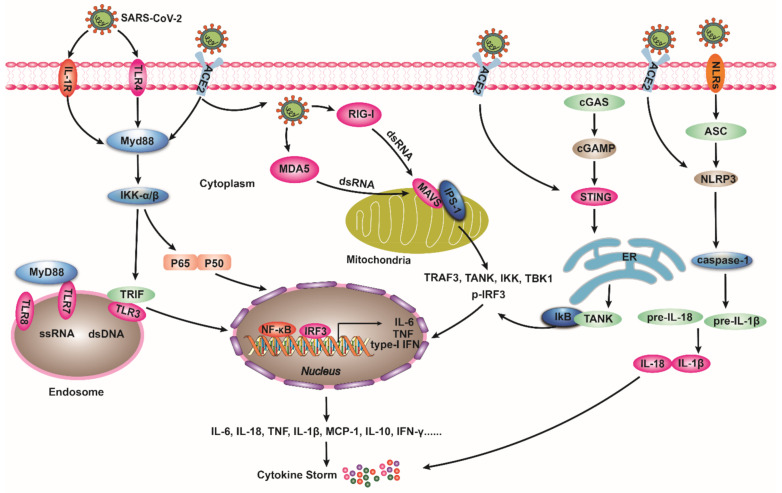
The major signal pathways of immunity against SARS-CoV2 infection. (i). TLRs primarily activate the transcription of NF-κB through MyD88 and TRIF-dependent pathways, which leads to the secretion of inflammatory factors and type-I IFN. (ii). RIG-I and MDA5 recognize the dsRNA in the cytoplasm and activate the downstream mitochondrial antiviral-signaling protein (MAVS) and IPS-1. Immune response is induced, including the activation of NF-κB and IRF3 transcription. (iii). cGAS identifies the dsDNA of the virus, thus inducing the production of the second messenger cyclic guanosine monophosphate adenosine monophosphate (cGAMP), which binds to STING on the endoplasmic reticulum membrane to cause an inflammatory response. As a member of the NLRs family, NLRP3 activates caspase-1, resulting in the cleavage of pre-IL-1β and pre-IL-18 to generate IL-1β and IL-18 in the lung inflammatory response. Abbreviations: TLRs, Toll-like receptors; MDA5, melanoma differentiation-associated protein5.

## Data Availability

Not applicable.

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
