# Peer review of "ACE2 and Innate Immunity in the Regulation of SARS-CoV-2-Induced Acute Lung Injury: A Review"

_ijms, 2021, doi:10.3390/ijms222111483_

Round 1

Reviewer 1 Report

This review shows reports on an important topic in a concise form. It is well organized, however, I have a few concerns. The main is that at some points it is not objective e.g. section 5.1.  Chloroquine. I don't understand why it ends with a phrase: "However, as there has been no clinical trial of chloroquine on  SARS, whether it is helpful for SARS treatment and whether the mechanism is related to ACE2 still needs to be confirmed."

There were at least 3 clinical trials on COVID treatments with this drug (ClinicalTrials.gov). Moreover, NIH, FDA, and WHO cautioned against the use of chloroquine for COVID-19. Please do some search on this topic and broaden this section.

Minor comments:

  1. Abstract: "In the lung, innate immunity as a critical line of defense against pathogens, including SARS-CoV-2."- there is probably the word "act" missing
  2. Chapter 2: "The active site of the catalytic domain, the zinc metallopeptidase domain, shares 41.8% of sequence identity with ACE. Despite this similarity... " - it is not a big similarity. Please rewrite. Maybe: "although, there is some similarity..."
  3. The Abbreviations are shown 2 times (pages 3 and 9). Is it useful?

4. The last sentence on page 4 is too complex. Please rewrite.

  1. There is probably also a mistake in the title of Table 1. Moreover, as I suggested before the other hand should be shown regarding chloroquine.

Reviewer 2 Report

Thanks for the opportunity to review the manuscript by Qu and colleagues. In general, the manuscript looks interesting and well-conducted, but some issues should be attended to before consider accept it for publication.

Although the topic is exciting and opportune, I found it superficial in most sections, particularly in the ACE2 and SARS-CoV-2 in ALI (2.2 section). Since the authors focus on the title the importance of ACE2/SARS-CoV-2 in ALI, they should emphasize recent basic and clinical findings, some examples: PMID: 34427852, 34449302, 34463644, 34610856.

The introduction of Section 5 (The potential therapeutic role of ACE2 in COVID-19) also is scanty. May you find useful the paper in this area to complete this introductory paragraph.

Minor issues:

Figure 1: Please correct "Aagiotensinogen" by Angiotensinogen;  "Rein" by Renin; "Pro-inflammtion" by Pro-inflammation. In pro-liferation, delete the hyphen; "Anti-inflammtion" by Anti-inflammation.

In figure 2 (and along with the manuscript), correct IFN-1 by type-1 IFN; Correct "Cymphocytes" by Lymphocytes.

The three figures should correct spaces between words.

Table 2 should be located before the Conclusions section.

The abbreviations table is incomplete.

Author Response

Responds to the Reviewer 2:

Thank you very much for reviewing the manuscript entitled by “ACE2 and innate immunity in Regulation of SARS-CoV-2 Induced Acute Lung Injury: A Review” (#ijms.1408486). Those comments are all valuable and very helpful for revising and improving our paper. We have carefully revised the manuscript according to the reviewers’ comments and revised the format of the references. Besides the following changes, we have corrected some expression errors and added more references. Hopefully, we could have our article considered of publication in your journal. If there are any other corrections we need to change, please feel free to contact us. We have marked changes in manuscript in red.

Q1: Although the topic is exciting and opportune, I found it superficial in most sections, particularly in the ACE2 and SARS-CoV-2 in ALI (2.2 section). Since the authors focus on the title the importance of ACE2/SARS-CoV-2 in ALI, they should emphasize recent basic and clinical findings, some examples: PMID: 34427852, 34449302, 34463644, 34610856.

A: Thank you very much for excellent comments. As your suggested, we added the latest references of ACE2/SARS-CoV-2 in ALI in section 2.2. (Page 3).

Q2: The introduction of Section 5 (The potential therapeutic role of ACE2 in COVID-19) also is scanty. May you find useful the paper in this area to complete this introductory paragraph.

A: Thank you very much for the constructive comments. We expanded the introduction of section 5 and added useful references (PMID: 34636804, 32105632, 34628234, 32714335) (page 8).

Q3: Figure 1: Please correct "Aagiotensinogen" by Angiotensinogen; "Rein" by Renin; "Pro-inflammtion" by Pro-inflammation. In pro-liferation, delete the hyphen; "Anti-inflammtion" by Anti-inflammation.

A: Thank you. We apologized for our carelessness and corrected the mistakes (Page 2).

Q4: In figure 2 (and along with the manuscript), correct IFN-1 by type-1 IFN; Correct "Cymphocytes" by Lymphocytes.

A: We are grateful to the reviewer for pointing out our error. In Figure2, we corrected “IFN-1” by “type-I IFN” and corrected “Cymphocytes” by “Lymphocytes” (Page 5).

Q5: The three figures should correct spaces between words.

A: Thank you very much for helpful comments. We have corrected the three figures (Pages 2,5,6).

Q6: Table1 should be located before the Conclusions section.

A: Thank you very much for the constructive comments. We corrected the position of Table 1 (Page10).

Q7: The abbreviations table is incomplete.

A: Thank you very much for excellent comments. We added the abbreviations in table (Page10).

Round 2

Reviewer 1 Report

The manuscript has been much improved, however, there is still something missing in the title of Table 1. The phrase:  "and in" is misleading.

Author Response

Q: The manuscript has been much improved, however, there is still something missing in the title of Table 1. The phrase: "and in" is misleading.

A: Thank you very much for helpful comments. We revised the title of Table1 to “The potential therapeutic role of ACE2 in SARS-CoV-2 induced acute lung injury”. We have marked changes in manuscript in red (Page10).